# The AwHog1 Transcription Factor Influences the Osmotic Stress Response, Mycelium Growth, OTA Production, and Pathogenicity in *Aspergillus westerdijkiae* fc-1

**DOI:** 10.3390/toxins15070432

**Published:** 2023-06-30

**Authors:** Yufei Wang, Fei Liu, Jingying Pei, Hao Yan, Yan Wang

**Affiliations:** 1College of Food Science and Technology, Zhejiang University of Technology, No. 18 Chaowang Road, Gongshu District, Hangzhou 310014, China; 221122260062@zjut.edu.cn (Y.W.); pjy_123456789@163.com (J.P.); 2Institute of Food Science and Technology, Chinese Academy of Agricultural Sciences, No. 2 Yuanmingyuan West Road, Haidian District, Beijing 100193, China; liufei0823@yeah.net; 3Zhejiang Provincial Center for Disease Control and Prevention, No. 3399 Binsheng Road, Binjiang District, Hangzhou 310051, China

**Keywords:** transcription factor AwHog1, ochratoxin A, osmotic response, *Aspergillus westerdijkiae*

## Abstract

*Aspergillus westerdijkiae*, known as the major ochratoxin A (OTA) producer, usually occurs on agricultural crops, fruits, and dry-cured meats. Microorganisms produce OTA to adapt to the high osmotic pressure environment that is generated during food processing and storage. To investigate the relationship between OTA biosynthesis and the high osmolarity glycerol (HOG) pathway, the transcription factor *AwHog1* gene in *A. westerdijkiae* was functionally characterised by means of a loss-of-function mutant. Our findings demonstrated that the growth and OTA production of a mutant lacking *AwHog1* decreased significantly and was more sensitive to high osmotic media. The *ΔAwHog1* mutant displayed a lower growth rate and a 73.16% reduction in OTA production in the wheat medium compared to the wild type. After three days of culture, the growth rate of the *ΔAwHog1* mutant in medium with 60 g/L NaCl and 150 g/L glucose was slowed down 19.57% and 13.21%, respectively. Additionally, the expression of OTA biosynthesis genes was significantly reduced by the deletion of the *AwHog1* gene. The infection ability of the *ΔAwHog1* mutant was decreased, and the scab diameter of the pear was 6% smaller than that of the wild type. These data revealed that transcription factor AwHog1 plays a key role in the osmotic response, growth, OTA production, and pathogenicity in *A. westerdijkiae*.

## 1. Introduction

Ochratoxin A (OTA), a widespread nephrotoxic mycotoxin, is frequently detected in many grains such as corn, soybeans, and wheat, as well as in products such as coffee, tea, wine, and bacon [1,2,3]. It has been shown that OTA has abundant toxicological effects, such as nephrotoxicity, hepatotoxicity, genotoxicity, teratogenicity, immunotoxicity, and carcinogenicity [4,5,6]. The International Agency for Research on Cancer has categorized OTA as a Group 2B carcinogen due to its substantial toxicity to both people and animals [6].

OTA is produced by filamentous fungi including *Aspergillus* and *Penicillium* spp. [1,7]. *Aspergillus westerdijkiae*, known as the major OTA-producing species, is a filamentous fungus derived from the *A. westerdijkiae* taxon. About approximately 70% of these strains are capable of producing OTA and are commonly found in foods such as coffee, milk, wine, beer, grapes, oranges, fruit juices, and dry-cured meats [7,8].

It has been discovered that the high osmotic glycerol (HOG) pathway regulates OTA production in *Penicillium* and *Aspergillus* in response to high osmotic conditions [9]. The biosynthesis of OTA plays an adaptive role in this habitat, as it has been shown previously that *P. nordicum* is adapted to NaCl-rich environments, such as dry-cured foods or even saline environments [10,11]. In *P. nordicum*, elevated concentrations of NaCl lead to high osmotic stress and subsequently promote the biosynthesis of ochratoxin A. The growth rates of *A. westerdijkiae* were probably higher in media supplemented with specific NaCl concentrations (0–80 g/L), and the maximum colony diameter was recorded when the concentration of NaCl was 40 g/L. The ability to produce OTA was inhibited at salinities of 40 g/L and 60 g/L in *A. westerdijkiae*. *A. westerdijkiae*’s ability to produce mycelium and spores significantly increased at a glucose concentration of 250 g/L. The production of OTA did not change between 20 and 100 g/L glucose concentration but decreased when the concentration reached 150 g/L [12].

Fungi have many metabolic pathways that help overcome extreme stress [13,14,15]. The HOG pathway is activated by increasing the ambient osmotic pressure and leads to an increase in cell glycerine concentrations to adapt to intracellular osmotic pressure [13,16]. The *hog1* gene is the primary gene in the high-osmolarity glycerol response pathway, which enables fungi to survive osmolarity and oxidative stress [17]. HOG is a mitogen-activated protein kinase (MAPK). The phosphorylation of Hog1 stimulates the expression of downstream genes, thus enabling fungi to resist various environmental pressures [18]. The HOG pathway is implicated in the regulation of growth, development, pathogenicity, fungicidal susceptibility, and responses to osmotic stress in various kinds of filamentous fungi [19]. Through the HOG signalling cascade, the high osmolarity of the environment is normally transmitted to the transcription level of downstream regulatory genes [18,20].

The same point of view found that the HOG pathway has an important role in maintaining fungal growth and that activation of the HOG pathway could be involved in regulating growth defects due to disruption of complex sphingolipid biosynthesis [21]. A recent report indicated that mutants lacking PtcB and CDC14, which were constructed in Hog1 phosphatases, exhibited slow growth, altered sporulation, and decreased virulence [22]. It has been shown that liamocin biosynthesis in the gene deletion strains was reduced by 98.09% when the key gene in the Hog1 signalling pathway, *hog1*, was deleted [23]. The previous study has shown that the correlation between NaCl concentration and ochratoxin A biosynthesis is associated with the HOG MAP kinase signalling cascade pathway [11]. The changing levels of the Hog1 signalling pathway have been studied in mycotoxin-producing fungi, such as *P.verrucosum* [24], *Fusarium proliferatum* [25], *F. graminearum* [26], *A. flavus* [27], and *Alternaria alternate* [7].

Based on the aforementioned circumstances, the topic of whether HOG transcription factors can help control the virulence and capacity for OTA biosynthesis of *A. westerdijkiae* was brought up. We concentrate on AwHog1, a possible transcription factor in *A. westerdijkiae*, in order to better comprehend the function of the HOG pathway. This study constructed a mutant with a Hog1 homologs gene deleted to show whether the *AwHog1* transcription factor could influence mycelium growth, spore production, OTA biosynthesis, responsiveness to high osmotic stress, and the infection ability of *A. westerdijkiae*.

## 2. Results

### 2.1. Characterization of a Hog1 Homology

MEGA6.0 software was used to construct a phylogenetic tree of Hog1 homologs based on sequences from 11 fungal species (Figure 1). As shown, Hog1 of *Aspergillus* spp. And *Penicillium* spp. Were highly homologous on the phylogenetic tree. The predicted amino acid sequence of *AwHog1* showed 79% homology with that of Hog1 (protein identification accession No. PLB43561.1) from *A. steynii*. However, the AwHog1 protein exhibited larger differences compared with Hog1 from *Fusarium* spp. and *Saccharomyces* spp.

### 2.2. Construction of ΔAwHog1 Deletion Mutant

To examine *AwHog1*’s role within *A. westerdijkiae*, *AwHog1* deletion mutants were generated through homologous recombination (Figure 2A). Fusion fragments of upstream, downstream, and hygromycin-resistant fragments were obtained and their composition was validated. A null fragment of the *AwHog1* gene was detected in *ΔAwHog1* mutant strains, indicating that the *AwHog1* gene has been replaced by a fusion fragment.

In order to examine how OTA production and colony expansion are affected by high osmotic circumstances, 1.7 M NaCl or 1.5 M glucose was added to the medium. The morphology of the colonies was observed for 84 h after incubation (Figure 2B). It was clear that the strain growing rates were obviously influenced by high osmotic conditions. A certain concentration of glucose promoted the growth of *A. westerdijkiae* fc-1. With the addition of 1.5 M glucose, colonies were larger in diameter than those in a non-glucose PDA medium. Conversely, the growth of *A. westerdijkiae* fc-1 was weakened when a certain concentration of NaCl was added into the medium. When 1.7 M NaCl was added, the diameter of the colony decreased compared to the control without NaCl. The OTA content of the *ΔAwHog1* mutant grown on the wheat medium was determined to assess whether *ΔAwHog1* deletion affects the production of OTA (Figure 2C). For the production of large amounts of OTA in *A. westerdijkiae* fc-1, the wheat medium is suitable. Three replicate experiments were carried out and toxin production rates of 533.5 ± 72.95 μg/g and 143.16 ± 33.01 μg/g were obtained for the wild type and *ΔAwHog1* mutant, respectively. The characteristic peaks of the liquid chromatogram can determine that the substance is OTA (Figure 2D,E). It was obvious that the deletion mutant strain *ΔAwHog1* produced lower toxin rates compared to that of the wild type, and OTA production decreased by 73.16% in the *ΔAwHog1* mutant.

### 2.3. High Concentrations of NaCl or Glucose Affect Colony Morphology

In order to examine the effects of high permeability conditions on *A. westerdijkiae*, Add 0, 20, 30, 60, 80, 100 g/L NaCl or 0, 50, 100, 150, 200, 250 g/L glucose to the medium. After 24 to 48 h incubation, observe the colony morphology. No significant changes in colony morphology or colour were found under high osmotic conditions (Figure 3).

The growth rate of mycelium is obviously affected by high permeability conditions, and the tendency of mycelium growth is similar between the *ΔAwHog1* mutant and the wild type. As the concentration of NaCl and glucose increases, the growth of mycelium tends to strengthen first and then weaken. Low concentrations of NaCl (20–80 g/L) promoted the growth of *A. westerdijkiae* fc-1. High concentrations of NaCl (more than 80 g/L) decreased the growth of *A. westerdijkiae* fc-1 (Figure 3A). When the concentration of NaCl is less than 60 g/L, the colony diameter of the wild type was 37% larger than that without NaCl, and the colony diameter of *ΔAwHog1* was only 16% larger than that without NaCl (Figure 3C). Glucose could promote the growth of *A. westerdijkiae* fc-1, and the growth of *A. westerdijkiae* fc-1 was slightly weakened due to the deletion of the gene *AwHog1* (Figure 3B). The diameter of the colonies was 61.4% and 55.4% larger than that without 150 g/L glucose in the wild type and *ΔAwHog1*, respectively (Figure 3C).

### 2.4. Growth Diameter, Spores Number, and OTA Production Are Affected and Modulated by AwHog1

To uncover the role of *AwHog1*, the growth, spore numbers, and OTA production under different concentrations of NaCl and glucose were compared. When the concentration of NaCl was 0–40 g/L, the mycelium growth diameters of the wild type and the *ΔAwHog1* mutant showed a gradual increase, and the growth diameters of both strains declined as the NaCl concentration increased (Figure 4A). The growth diameters of the wild type and the *ΔAwHog1* mutant increased when the concentration of glucose was 0–150 g/L, and with the increase in glucose concentration, the growth diameters of *ΔAwHog1* mutant and the wild type decreased rapidly (Figure 4B). Compared with the wild type, the growth diameter of the *ΔAwHog1* mutant under different concentrations of NaCl and glucose was lower than that of the wild type (Figure 4A,B), indicating that the deletion strain *ΔAwHog1* would slow the growth of *A. westerdijkiae* fc-1. The spore numbers of the wild type and *ΔAwHog1* increased when the concentration of NaCl was 0–40 g/L, then both strains decreased with the increase of NaCl concentration (Figure 4C). The spore numbers of *ΔAwHog1* were higher than those of the wild type at NaCl concentration of less than 20 g/L. When the concentration of NaCl was between 40 and 100 g/L, the spore number of *ΔAwHog1* was lower than the wild type. The result, which was similar to a previous report [12], showed that the spore number of the wild type and *ΔAwHog1* increased overall when the concentration of glucose was 0–150 g/L (Figure 4D). When the concentration increased to 200 g/L glucose, the spore numbers of the wild type and *ΔAwHog1* decreased. The spore number of *ΔAwHog1* was higher than the wild type at glucose concentrations less than 150 g/L (Figure 4D). The OTA content in the deletion mutant strain *ΔAwHog1* was almost undetectable in PDA or a YES plate (Figure 4E,F). The OTA production of the wild type and *ΔAwHog1* increased when the concentration of NaCl was 0–20 g/L and then decreased with the increase of NaCl concentrations (Figure 4E). The OTA production in the *ΔAwHog1* mutant grew rapidly at glucose concentrations of 100 and 150 g/L (Figure 4F). Similar to the earlier report [12], the OTA production of *ΔAwHog1* under different concentrations of NaCl and glucose was lower than that of the wild type.

### 2.5. Glycerol Content Is Affected by AwHog1

To evaluate whether the absence of AwHog1 affects the glycerol content, the glycerol content of the *ΔAwHog1* mutant and the wild type grown in different concentrations of NaCl was detected. The glycerol content of the *ΔAwHog1* mutant was smaller than that of the wild type when the concentration of NaCl was between 0 and 20 g/L. The trend of the wild type and the *ΔAwHog1* mutant increased with the increase of the NaCl concentration and then decreased when the concentration rose to 70 g/L. The *ΔAwHog1* mutant has a higher glycerol content than the wild type under conditions of high NaCl concentration (Figure 5).

### 2.6. Relative Expression of AwHog1 and OTA Biosynthetic Genes

The expression of *AwHog1* and OTA biosynthetic genes under different concentrations of NaCl was detected by PCR. Compared with the wild type, the *AwHog1* gene was not expressed in the *ΔAwHog1* mutant at different NaCl concentrations (Figure 6A). The expression of OTA biosynthetic genes tended to initially increase and then decrease with the increase in the concentration of NaCl, and the phenomenon was similar to the level of the *AwHog1* gene (Figure 6). The relative gene expression of *otaA*, *otaB,* and *otaR1* in the *ΔAwHog1* mutant was lower than that of the wild type in the media without NaCl (Figure 6B–D). The difference was more obviously affected by the deletion of AwHog1 in the medium containing 20 g/L NaCl. The OTA content was not detectable in *ΔAwHog1* mutant without salt addition, and the OTA accumulation was the largest in the *ΔAwHog1* mutant and the wild type under 20 g/L NaCl. The tendency of OTA biosynthetic gene expression was similar to OTA accumulation changes.

### 2.7. Transcription Factor AwHog1 in A. westerdijkiae fc-1 Is Essential for Fungi Infection

*A. westerdijkiae* fc-1 and the *ΔAwHog1* mutant were inoculated into pears to determine whether AwHog1 plays a role in the ability to infect. At 6 days after infection, the diameter of the lesion was measured (Figure 7). After 6 days of incubation, the scab diameter of the wild type-infected pear was 3.32 ± 0.09 cm, while the scab diameter of the *ΔAwHog1* mutant-infected pear was 3.13 ± 0.02 cm. The scab diameter of the infection from the *ΔAwHog1* mutant is about 6% smaller than that of wild type, which demonstrates that *AwHog1* actively regulates *A. westerdijkiae*’s capacity for infection.

## 3. Discussion

In this study, we have successfully constructed a *ΔAwHog1* mutant. Then, by analysing the effects of *AwHog1* deletion on colony morphology, mycelium growth, spore generation, toxin accumulation under osmotic stress, and the infection capacity of *A. westerdijkiae*, the role of the transcription factor *AwHog1* was identified. We discovered that the *ΔAwHog1* mutant had growth flaws, a reduction in the amount of ochratoxin A produced on various mediums, increased sensitivity to osmotic stress, and impaired infection capacity. These might be connected to the HOG pathway that the *Hog1* gene activates.

The expression of the *Hog1* gene encoding transcription factor can regulate the adaptive response to osmotic stress and can mediate the responses induced by osmotic stress [28,29]. In *Sclerotinia sclerotiorum*, the increased sensitivity to NaCl and glucose media was observed in *ΔAwHog1* mutants, which is in line with previous research findings that Hog1 mutants were more sensitive to sugars and NaCl [30]. The members involved in the HOG pathway show the characteristics of adapting to the osmotic stress source in filamentous fungi. There was an increased sensitivity to hyperosmolarity in the Hog1 mutant of *M. oryzae*. The elements SKN7 and SsK1 in the HOG pathway have been shown to have a role in the effects of osmotic stress [31]. Meanwhile, the *Hog1* gene could influence mycelium growth and spore number. The *bos5* gene was identified as a key component of the HOG pathway. It has been found that the deletion of the *bos5* gene in *B. cinerea* would result in a significant decrease in the growth rate and spore yield of the aerial mycelium [32]. The *StPBS2* gene was found to be involved in the HOG pathway mechanism. In comparison with the wild type strain, the *StPBS2* gene showed decreased colony growth in *Setosphaeria turcica* [33].

The Hog signal pathway plays a major role in the biosynthesis of toxins and accumulation in filamentous fungi. The transcriptomics was applied to study OTA biosynthetic expression genes and stress-related genes in *P. nordicum* strains [34]. *P. nordicum* is able to successfully adapt to the high osmolarity stress caused by the production of up to 22% NaCl in the manufacture of cured meats such as sausages and hams [35]. *FgPrb1* is involved in the osmotic stress response of *Fusarium graminearum* via the HOG pathway. It has been shown that DON biosynthesis and the pathogenicity of *F. graminearum* were greatly reduced when *FgPrb1* was deleted [36]. The quantitative PCR outcomes of this investigation showed that the expression of OTA biosynthetic genes in the *ΔAwHog1* mutant was lower than that of the wild type. We found that compared with the wild type, deletion of the *AwHog1* gene would reduce the mycelium growth rate, spore number, and OTA accumulation. 

The HOG pathway is particularly important in pathogenic fungi because of its traits related to pathogenicity. The inactivation of HOG pathway components could decrease the pathogenicity of fungi. These include biofilm formation, surface adhesion, morphological occurrence, and epigenetic transformation [37]. The STK1 was found to be highly homologous with the HOG MAPK gene *HOG1* of *Saccharomyces cerevisiae* and it had diminished virulence, a significantly altered toxin, and reduced pathogenicity on the leaves of susceptible inbred corn OH43 [38]. Msn2, a general HOG pathway stress response transcription factor, plays pleiotropic roles in infection-related morphogenesis and pathogenicity in pathogenic fungi [39]. The mutants *Δpgpbs1* and *Δpgpbs* showed a decrease in pathogenicity during experimental induction of infection of barley seeds and leaves, indicating that pgpbs1 and pgpbs play key roles in the HOG signalling pathway of *P. graminea*. [19]. In our study, the diameter of the infection of *ΔAwHog1* mutant was about 6% smaller compared with the wild type. As a result, the HOG pathway genes have been shown to improve resistance to some fungi.

## 4. Conclusions

According to the results, the lack of *AwHog1* has a negative effect on colony morphology, mycelial growth, spore production, OTA accumulation, and the infection ability of *A. westerdijkiae*, while increasing the susceptibility of the mutant to high osmotic pressure environments. Therefore, we believe that the *AwHog1* transcription factor plays a key role in influencing *A. westerdijkiae* growth, hyperosmotic response, OTA biosynthesis, and pathogenicity by regulating HOG pathway.

## 5. Materials and Methods

### 5.1. Strains and Media

In this study, the wild type strain *A. westerdijkiae* fc-1 was isolated and characterized in our laboratory [5]. The strains were usually cultured in the dark at 28 °C. Conidia were scraped from a PDA plate with a sterile cotton swab, cultured for four to eight days, and then resuspended in sterile water. A blood cell counter was used to adjust the number of conidia to 1.0~2.0 × 10^7^ conidia/mL. Conidial suspensions of the wild type and mutant strains were stored at −80 °C with 15% glycerol.

### 5.2. Culture Conditions

The OTA-producing strain *A. westerdijkiae* fc-1 was put in potato glucose agar (PDA; potato 200 g/L, glucose 20 g/L, agar 20 g/L) [12] for 7 days. The *ΔAwHog1* mutant strain was built and stored in our lab. To produce ochratoxin A, the culture was grown and cultured in yeast extract sucrose agar [12] (YES; Yeast extract 20 g/L, sucrose 150 g/L, agar 15 g/L) with different concentrations of NaCl (0, 20, 40, 60, 80 and 100 g/L) and malt extract-agar (MEA) medium supplemented with glucose in different amounts (0, 50, 100, 150, 200 and 250 g/L). Scraping the PDA plate with a sterile cotton swab was used to obtain the best viable conidia on the 5th day of culture and 10 mL sterile water containing 0.01% Tween 80 was used to collect the spores; spores were counted with a hemocytometer, adjusted to 10^7^ conidia mL^−1^ [40], and used as an inoculum. The spore suspension was preserved in glycerine solution at −80 °C. Each agar plate was inoculated with 5 μL of spore suspension and incubated at 28 °C for 7 days. All experiments were repeated 3 times, with 3 treatments each time.

### 5.3. Construction of ΔAwHog1 Deletion Strains

In order to obtain the mycelia for DNA extraction, 10 μL of conidial suspension was inoculated into 20 mL liquid potato glucose broth (PDB) and incubated at 180 r/min [40] and 28 °C. After 84 h of incubation, we collected and dried the mycelium. The mycelium was precooled in liquid nitrogen and stored at −80 °C.

Two pairs of primers were used to detect the insertion of the *AwHog1* gene selection marker and the deletion of the *AwHog1* gene (Table 1). In order to construct *AwHog1*-missing strains, primers *AwHog1*-up-F/R and *AwHog1*-down-F/R were respectively used to clone the 1.4 Kb upstream and downstream DNA fragments. The hygromycin B phosphotransferase (hyg) gene from the pCAMBIA-1300 vector (Abcam, Cambridge, UK) was cloned using the hyg-F/R primer combination as a resistance marker. The upstream/downstream sequences, hyg gene cassette, and knockout primers *AwHog1*-knock-F/R were used to obtain a replacement cassette of *AwHog1* through nested PCR. *A. westerdijkiae* protoplasts were created from purified PCR fusion products using the SanPrep column DNA gel extraction kit (SANGON). The preparation of the protoplast and the transformation process were carried out in accordance with the previous work. A total of 300 μL of the transformed product was inoculated into a HYG resistant plate with 100 μg/mL hygromycin B. Each mutant was repeated with ten plates. An ordinary PDA board served as a positive control. A single colony that had been incubating for 3 to 4 days was collected for identification. The specific primers *AwHog1*-in-F/R and *AwHog1*-out-F/R were used to identify the *ΔAwHog1* mutant. The specific primers *AwHog1*-in-F/R were used to amplify the internal genes of *AwHog1* from the *A. westerdijkiae* genomic DNA by PCR (Table 1).

### 5.4. OTA Production Analysis in the ΔAwHog1 Mutants and the Wild Type

In order to investigate the production of OTA, 5 mL of conidial suspension (wild type and *ΔAwHog1* mutant) was placed in a 100 mL Ellen Meyer flask containing 25 g of wheat. The initial water activity of wheat was modified to 0.9 from 0.35 by adding 15% water. After incubating at 28 °C for 4 days [41], 5 g wheat culture was added to 25 mL methanol and agitated for 1 min. After centrifugation at 7500 g, the supernatant was separated. The supernatant was then put into a vial after being passed through a 0.22 m filter. Then, a C_18_ column (4.6 mm × 150 mm, Agilent, Santa Clara, CA, USA) [42] was used, and 20 μL of each sample was injected into an HPLC-FLD consisting of Agilent 1260 infinity, using 330 nm for the excitation wavelength and 460 nm for the fluorescence detector. The mobile phase’s main component was acetonitrile: water: acetic acid (99:99:2 *v*/*v*/*v*); column temperature was 37 °C, and the flow rate was 1 mL/min.

### 5.5. Mycelial Growth, Conidia Count, and OTA Production

To analyse the colony diameter, 5 μL suspension of 10^7^ conidia mL^−1^ was positioned in the centre of a solid medium drop by drop. The cross method was used to measure the diameter of the colonies growing every 12 h. Each group was treated with either NaCl or glucose. Colony morphology and colour were observed during culture, and mycelium morphology was observed under a microscope on the 4th day of culture.

Five agar plugs (8 mm in diameter) were removed from the colony and put into 10 mL microreactive tubes with 5 mL sterile water that contained 0.01% Tween 80. The spores were shaken for two hours at room temperature on a rotary shaker, and a hemostatic meter was used to count the conidia (Fisher Scientific Company, Loughborough, UK). All experiments were performed three times for each treatment and repeated twice.

In order to determine the production of ochratoxin A, the strain was incubated on agar plates supplemented with NaCl or glucose at 28 °C. Five 8 mm-diameter agar plugs were removed from the colony and placed in a 2 mL micro reaction tube, and then 1 mL methanol was added. The mycelium of the fungus was extracted for 2 h at room temperature on a rotary shaker; the mycelium was discarded, and OTA was immediately identified by HPLC-FLD after the supernatant was filtered through a 0.22 μm filter and placed in a brown vial.

### 5.6. Phenotypic Comparison of Wild Type and ΔAwHog1 Mutants in Different High Permeability Conditions

The conidial suspensions (2.5 μL, 10^7^ conidia mL^−1^) of the wild type strain and *ΔAwHog1* mutant were added dropwise to the middle of each plate. Five replicates of each experiment were performed. The culture was incubated in the dark at 28 °C. The growth rate was assessed by examining the colony diameter of the wild type and *ΔAwHog1* mutant under different NaCl and glucose conditions every 12 h with the cross-sectional method. The formula below was used to compute the percentage of mycelial growth inhibition: inhibition rate (%) = [(C − T)/C] × 100%, where T was the average colony diameter of the test group given either NaCl or glucose treatment, and C was the average colony diameter of the control group. 

### 5.7. Glycerol Content of Wild Type and ΔAwHog1 Mutant on PDA Medium with Increased NaCl Concentration

Measurements were made of the glycerol content of the wild type and *AwHog1* mutant cells cultured on PDA medium with the addition of NaCl. The glycerol assay kit (purchased at the Institute of Bioengineering in Nanjing) employed the GPO Trinder enzymes and the glycerol phosphatase technique to measure the glycerol content of liquid samples using colorimetry. The glycerol concentration was measured according to the formula: Glycerol concentration = [Calibration A/Sample A] × calibration concentration (mmol/L).

### 5.8. Gene Expression under Various NaCl Environments

The wild type and *ΔAwHog1* mutant were cultivated under three distinct NaCl concentrations (0, 20, 70 g/L) to learn how OTA biosynthesis genes are expressed and controlled under various high permeability conditions. Each plate was placed on the same size of cellophane, which is a semi-permeable membrane. The conidial suspensions (200 μL) were inoculated with the wild type and *ΔAwHog1* mutants in the centre of the plate. After being cultured at 28 °C for 84 h in the dark, they were discarded. The cultures were collected and immediately precooled in liquid nitrogen and then stored at −80 °C prior to use. The RNA Easy Plant Mini Kit (QIAGEN, Hilden, Germany) was used to extract total RNA. Using an RNAPCR kit (AMV) (TAKARA, Omatsu, Shiga, Japan), reverse transcription PCR (RT-qPCR) was carried out, and the specific primers used are shown in Table 2. The cDNA and primers were mixed with the SYBR green PCR master mixture and RT-qPCR was performed in the 7500 real-time PCR system (Applied Biosystems, Foster City, Foster City, CA, USA). The relative expression of the gene was calculated using the internal control group GADPH, and the relative quantitative expression of the target gene was calculated with the 2^−ΔΔCT^ method, where ∆∆CT = (CT, Target—CT, Gadph) Treatment—(CT, Target—CT, Gadph) Control.

### 5.9. Infection Ability in Pears

Three pears (*Pyrus* × *bretschneideri*) of uniform size without wounds were chosen to test the infection ability of the *ΔAwHog1* mutant. The pears were washed three times with 0.01% sodium hypochlorite and then dried. Each pear was pricked with a sterile needle to make a wound (1 mm in diameter and 3–4 mm in depth). Then, with sterile water and OTA standard solution as controls, 10 µL of spore suspension of the wild type and deletion mutant was dropped on the wound. After culturing in the dark at 28 °C for 6 days, the diameter of the colony was measured using the cross-section method and photographs were taken. The conidia suspensions of the wild type and *ΔAwHog1* deletion mutant were injected with 10 μL sterile water, respectively, and the OTA standard was used as a control.

### 5.10. Statistical Analysis

SPSS Statistics 21.0 (Chicago, IL, USA) and Microsoft Excel 2019 (Redmond, WA, USA) were used to conduct all statistical analyses. Analyses of gene expression were evaluated using one-way analysis of variance (ANOVA). The mean value comparison was analysed by the Tukey procedure. The difference was considered significant at *p* < 0.05.

## Figures and Tables

**Figure 1 toxins-15-00432-f001:**
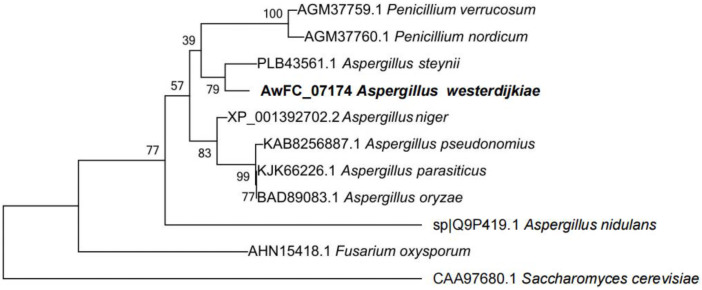
Phylogenetic tree of AwHog1 in *A. westerdijkiae* fc-1 with 11 fungal Hog homologs genes.

**Figure 2 toxins-15-00432-f002:**
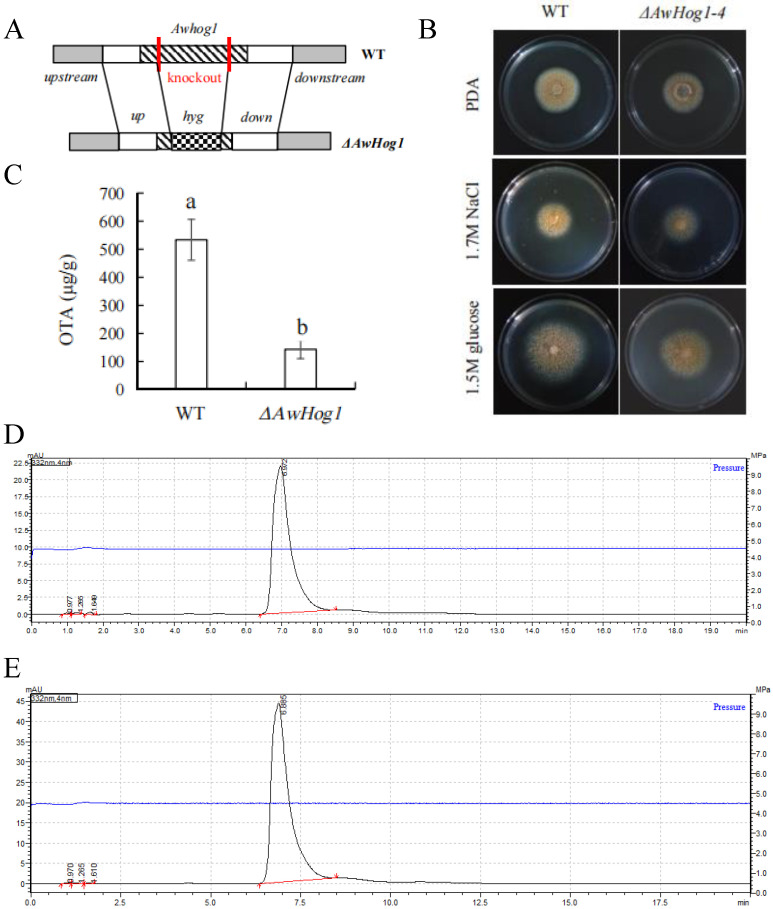
Construction of the *ΔAwHog1* mutant and characteristic analysis. (**A**) Construction of the *ΔAwHog1* mutant, (**B**) Colonies of the *ΔAwHog1* mutant and the wild type grown on PDA plates at 1.5 M glucose, 1.7 M NaCl, and PDA medium, which was used as a control. Strains were incubated at 28 °C for 6 days and photographed. (**C**) OTA content of the wild type and the *ΔAwHog1* mutant grown on wheat medium. Chromatograms of different concentrations of OTA detected in the wild type (**D**) and the *ΔAwHog1* mutant (**E**). Different letters indicate a significant difference between the corresponding values (*p* < 0.05).

**Figure 3 toxins-15-00432-f003:**
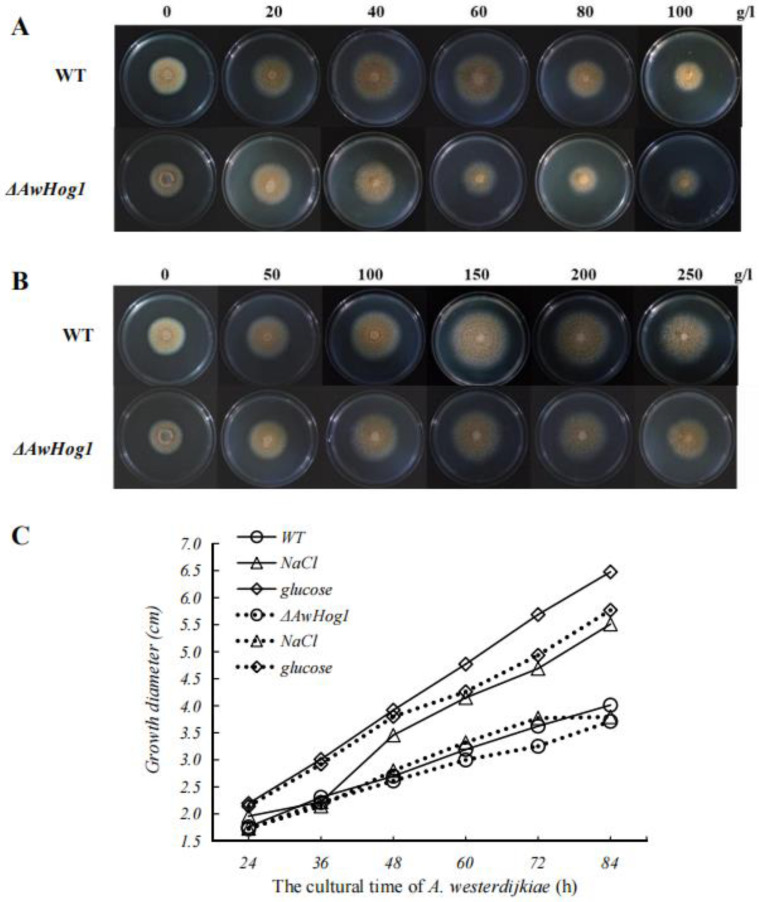
The colony of the *ΔAwHog1* mutant and the wild type after incubation at 28 °C for 84 h (**A**) Colonies of the wild type and the *ΔAwHog1* mutant grown on 0, 20, 40, 60, 80, 100 g/L NaCl-supplemented PDA plates. (**B**) The colony diameter of the *ΔAwHog1* mutant and the wild type grown on glucose-supplemented PDA plates at different cultivation times. (**C**) Colonies of *ΔAwHog1*(dotted line) and the wild type (full line) grown on media supplemented with 0 g/L NaCl (rotundity), 60 g/L NaCl (triangle), and 150 g/L glucose (lozenge).

**Figure 4 toxins-15-00432-f004:**
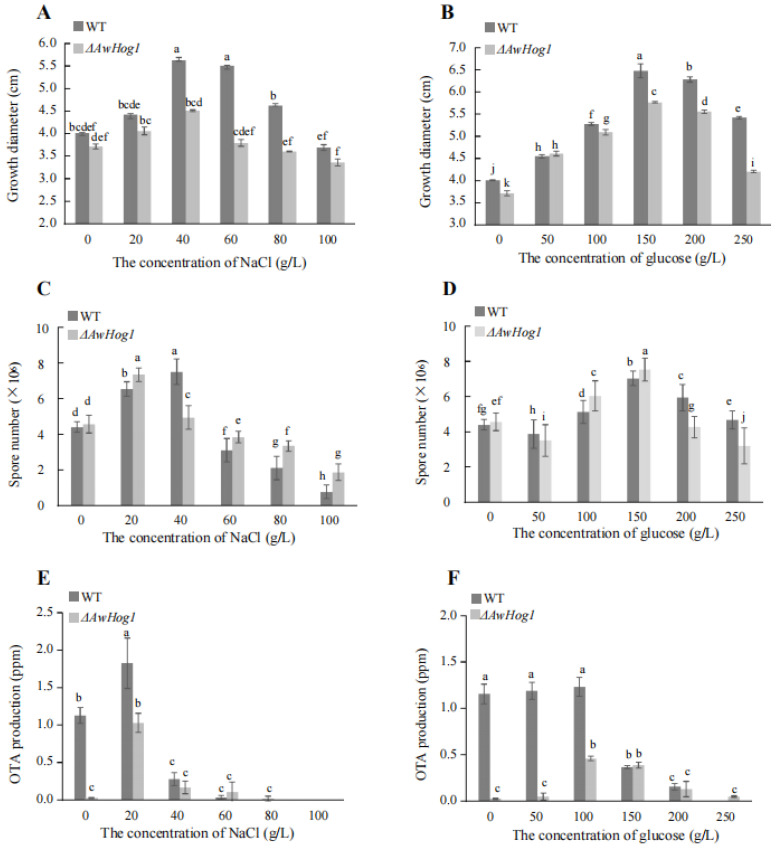
Response of the wild type and the *ΔAwHog1* mutant of *A. westerdijkiae* at different concentrations of NaCl and glucose. Colony growth diameters of the wild type and the *ΔAwHog1* mutant under different concentrations of NaCl (**A**) or glucose (**B**). Spore numbers of the *ΔAwHog1* mutant and the wild type under different concentrations of NaCl (**C**) or glucose (**D**). OTA production of the *ΔAwHog1* mutant and the wild type under different concentrations of NaCl (**E**) or glucose (**F**). Different letters indicate a significant difference between the corresponding values (*p* < 0.05).

**Figure 5 toxins-15-00432-f005:**
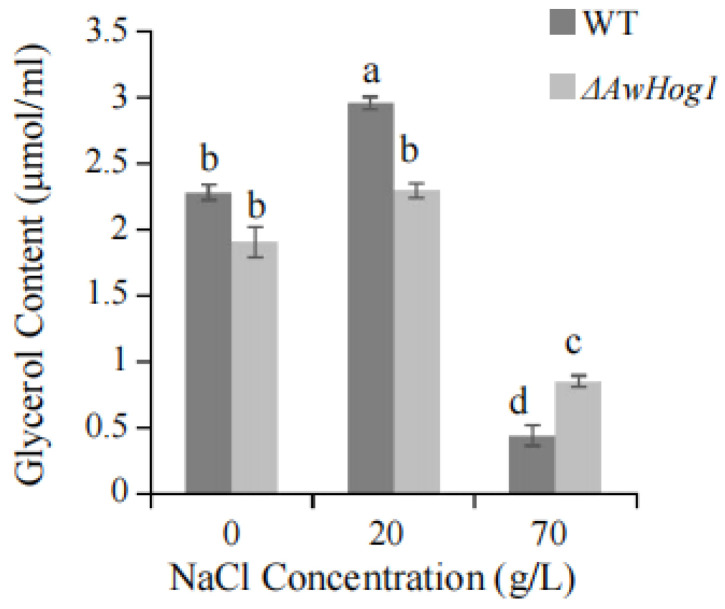
The glycerol content of the *ΔAwHog1* mutant and the wild type at different concentrations of NaCl. Different letters show a significant difference between the corresponding values (*p* < 0.05).

**Figure 6 toxins-15-00432-f006:**
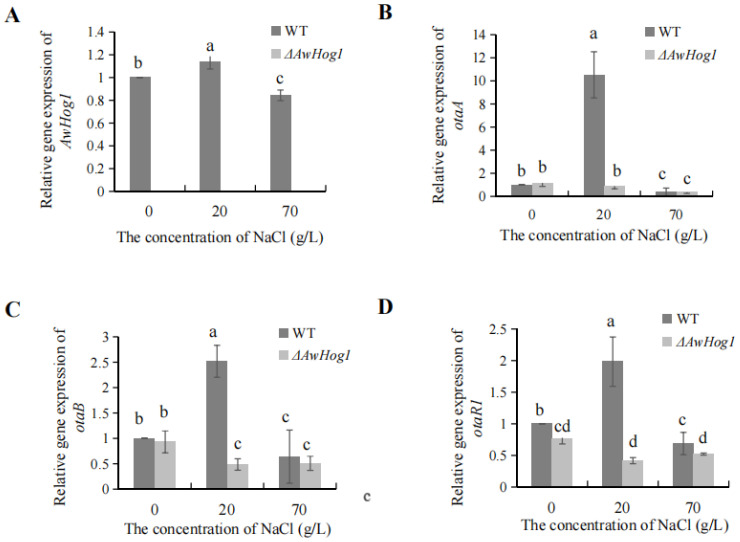
The relative levels of gene expression of *AwHog1* (**A**) and OTA biosynthetic genes *otaA* (**B**), *otaB* (**C**), and *otaR1* (**D**) under different concentrations of NaCl. Different letters show a significant difference between the corresponding values (*p* < 0.05).

**Figure 7 toxins-15-00432-f007:**
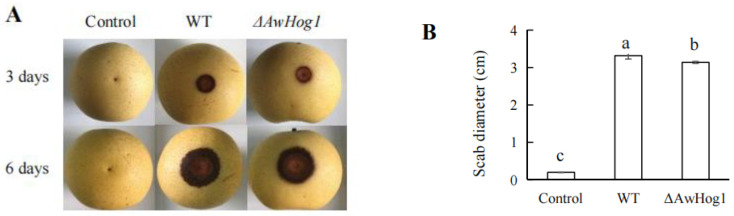
Pathogenicity assay for the *ΔAwHog1* mutant and the wild type strains in *A. westerdijkiae* fc-1 on pears (*Pyrus* × *bretschneideri*). (**A**) The infected pears were photographed after 6 days of 28 °C incubation. (**B**) The diameter of the scab was measured by the cross-section method. Control, pears are infected with the same amount of water instead of spore suspension. Different letters indicate a significant difference between the corresponding values (*p* < 0.05).

**Table 1 toxins-15-00432-t001:** Primers used for constructing *ΔAwHog1* cassettes and screening transformants.

Primer Name	Sequence (5′ to 3′)	Base (bp)	Product Size (bp)
*AwHog1*-Up	Forward—AGGAACATCTTGCATGGGCA	20	1425
	Reverse—CAAAATAGGCATTGATGTGTT GACCTCCCAGCCATACCAGACCAGTCC	48	
*AwHog1*-Down	Forward—CTCGTCCGAGGGCAAAGGAAT AGAGTAGACGATGCTGAGCTACCAGTG	48	1548
	Reverse—TCCTGGTGATACGGAGGGAG	20	
*AwHog1*-Knock	Forward—TTGGGGTTCTAGTCCGTGTC	20	3698
	Reverse—GGCAGTCACTTACCTCGCAT	20	
*AwHog1*-Out-F	Forward—TTGTGGAGACAAGGAAGCCG	20	860
	Reverse—TCGAAGCTGAAAGCACGAGA	20	
*AwHog1*-In-F	Forward—GCGTTTGGACTGGTCTGGTA	20	1795
	Reverse—TTGACATGGTCCTTTCCGGG	20	

**Table 2 toxins-15-00432-t002:** Primers used to amplify GADPH, otaA, otaB, otaR1, and *AwHog1* through real time PCR.

Primer Name	Sequence (5′ to 3′)
GADPH-F	CGGCAAGAAGGTTCAGTT
GADPH-R	CTCGTTGGTGGTGAAGAC
otaA-F	GGATCTTTATGACCGAATCAG
otaA-R	CCTTGACCTGAAGAATGCT
otaB-F	ATACCACCAGAGCTCCAAA
otaB-R	GAGATGTTCGGTCTGTTCA
otaR1-F	GCTTTCAAATCGAATGATTCC
otaR1-R	GATCGGTTGGAAGTGTAGAA
*AwHog1*-F	GGTCGCCGTCAAAAAGATTA
*AwHog1*-R	CAGCTCGGTCACGAAGTAGA

## Data Availability

Not applicable.

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
