# Peer review of "The AwHog1 Transcription Factor Influences the Osmotic Stress Response, Mycelium Growth, OTA Production, and Pathogenicity in Aspergillus westerdijkiae fc-1"

_toxins, 2023, doi:10.3390/toxins15070432_

Round 1
Reviewer 1 Report
The authors have descripted the pivotal roles of the AwHog1 transcription factor in ochra- 2 toxin A biosynthesis in Aspergillus westerdijkiae. The paper is interesting but more informations are necessary.
Line 34-35: please add more references ( DOI 10.3390/antiox10081239)
Line 43: please specify before the name and then the abbreviation
Line 56: please add references
Line 87-line 230: the methods used to obtained the results (for exemple table of primers) ishould be displaced in materials and methods, not in results
Line 232: the discussions need to be further explored. The results are unclear.
Line 284: indicate the paper in which do you have use this time of culture
Author Response
Thanks for your comments concerning our manuscript entitled "The AwHog1 transcription factor in ochratoxin A biosynthesis in Aspergillus westerdijkiae (toxins-2424369)". Those comments are valuable and helpful for us to revise and improve the paper. We have considered comments carefully and made some corrections. Revised portion are marked in blue in the manuscript.

Reviewer 2 Report
In the manuscript toxins-2424369, the authors constructed a ΔAwHog1 deletion mutant of Aspergillus westerdijkiae fc-1. Subsequently, the mutant was evaluated in terms of colony morphology, mycelium growth, spore production, infection ability, and ochratoxin A (OTA) production under osmotic stress. According to what was reported by the authors, the AwHog1 transcription factor plays key roles in the osmotic stress response, growth, OTA production, and infection ability. In general, this is an interesting manuscript with important implications; consequently, the work is worthy of publication in Toxins. However, several irregularities in the paper preclude acceptance in its present form. The following shortcomings stand out in the presented manuscript:
Title
Possible change to: The AwHog1 transcription factor influences the osmotic stress response, mycelium growth, OTA production, and pathogenicity in the fungus Aspergillus westerdijkiae fc-1
Abstract
L6-7. The sentence is rather confusing.
L8. High Osmolarity Glycerol (HOG) pathway
L14. Delete the word “supplement”
L18. Delete the word “regulation”
Introduction
I think the entire section needs to be rearranged and shortened. Overall, the study lacks any motive/objective. This section should be modified to describe the significance of this study.
L31-2, L43-5, L52-55, L67-9, L73-8, L79-82. The meaning of these sentences is not clear.
L60. Mould fungi?
Results
Figure 1. Deduced amino acid sequence alignment or phylogenetic tree?
Table 1. transformants
L106-7. The sentence is somewhat confusing.
Figure 2, profile C and Figure 4, profile E. Please add some representative chromatograms. Please revise the OTA contents!
Figure2, profile C. Statistical analysis is required.
Figure 3, profile C. Colony diameters at 96 h are missing!
L169 and 179. Similar to what report?
Figure 4, 5, 6, and 7. Statistical analysis is required.
L219-220. After 8 d of incubation…. The image (Figure 7, profile A) shows pears infected with the ΔAwHog1 mutant and the wild type strains at 3 and 6 d!
L222-24. The statistical analysis needs to be addressed!
Discussion
The discussion of the results is insufficient, and the author’s views need to be supplemented. It is probably due that you start with a general and detailed introduction before a connection to a minor result presentation. I am used to the opposite, which is perhaps more common. One starts with a short results presentation followed by reference to other publications and related discussion.
L258-60, 269-71. The sentences are rather confusing.
Conclusion
Please improve the conclusion section with clear findings.
Material and methods
Did the authors use any test to confirm the presence of the A. westerdijkiae fc-1 strain?
L288-9. The meaning of the sentence is not clear.
L295. Malt extract-agar (MEA) medium
L296-301. Repetitive text!
L306. Revolution is not a SI unit. Rotational frequency should be given in hertz (Hz), centrifugal force should be given in multiples of g (9.81 m/s2), and angular speed should be given in radians per second (rad/s).
L301. The concentration of the spore suspension lacked theoretic support! Spore viability should be definitively included in the document.
L308. precooled?
L331. DNA fragments
L312-14. The sentence is somewhat confusing.
L324. OTA production analysis. Please provide more technical details.
L327. wheat variety? samples of wheat were analyzed for OTA (and other mycotoxins) prior inoculation? Please add the initial and final moisture content of the inoculated wheat grain. Why did the author choose this incubation period?
L330-3. Please provide more details about the conditions of the HPLC methodology
L344-5, L354-8, L364-7, L371-5. The meaning of the sentences is not clear.
L388. How many pears? Variety? Were the fruits wounded at different points? Were the control fruits handled similarly? Was the experiment performed twice?
L391-5. Confusing text.
Statistical analysis
The Duncan`s multiple range test is a widely criticized statistical post-hoc test, because it has much less control over the type I error rate. Please use the Tukey procedure for means separation.
The English style and grammar are not satisfactory, and the language is not clearly written. The authors should consider using a scientific writing service to correct problems in English style and grammar.
Author Response

(The authors gave the same response as above.)

Reviewer 3 Report
1. What is the main question addressed by the research?
R: Functional characterization of the AwHog1 gene in A. westerdijkiae by means of a loss-of-function mutant
2. Do you consider the topic original or relevant in the field? Does it address a specific gap in the field?
R: It is of medium-average interest. Previous work was published by the author on the same research field.
3. What does it add to the subject area compared with other published material?
R: New mutant
4. What specific improvements should the authors consider regarding the methodology? What further controls should be considered?
R: No further experimental work needed
5. Are the conclusions consistent with the evidence and arguments presented and do they address the main question posed?
R: The aim of research is clear, and the methodological approach and implementation are correct. The discussion adequately arguments on the results, and conclusions support the main findings.
6. Are the references appropriate?
R: Yes, they provide the relative feedback / support the findings
7. Please include any additional comments on the tables and figures.
R: No further comments
No further comments
Author Response

(The authors gave the same response as above.)

Round 2
Reviewer 1 Report
The paper is ready to be published
Author Response
Thanks for your review again.
Reviewer 2 Report
The manuscript toxins-2424369 has been improved since the first submission. I have, however, still a few comments to the revised document.
L326. Again, revolution is not a SI unit.
L29. Conidia with the best viability… best viability? Spore viability should be included in the manuscript.
L325. Again, why did the author choose this incubation period?
L388. Please state the number of fruits!
On a less important note: The entire manuscript should be revised extensively particularly in its English, as some sentences are rather confusing.
Author Response
Thanks for yor review again.
The manuscript toxins-2424369 has been improved since the first submission. I have, however, still a few comments to the revised document.
L326. Again, revolution is not a SI unit.
I have modified it into "7500 g" in line 324
L291. Conidia with the best viability… best viability? Spore viability should be included in the manuscript.
L290. According to references [12], [40] and [42], the spore suspensions used in the experiments were obtained after 4-8 days of incubation, and we have chosen day 5 with reference to this time period. Usually, spore vitality is best when grown on PDA culture medium for 4-8 days. We have added this sentence “Scrape the PDA plate with a sterile cotton swab to obtain the best viable conidia on the 5th day of culture and 10 mL sterile water containing 0.01% Tween 80 was used to collect the spores, counte.” in Line 292-293.
L325. Again, why did the author choose this incubation period?
It was a mistake in last revison. The incubation time was amended to 4 days, on Line 325. Reference [41] was added, which showed that Ochratoxin A production was increased from day 3 to day 6, and peaked on day 6 of incubation on grain media and remained constant in later days. Here, 4 days of incubation was chosen to analyze OTA production.
- Liu, F.; Wang, Y.; Wang, L.; Wang, Q.; Wang, Y.; Liu, Y. Effect of different media on the growth and toxicity production capacity of Aspergillus ochraceus. J. Nucl. Agric. Sci. 2017, 31(4), 702-710.
L388. Please state the number of fruits!
3 pears were used in the experiment, and changes have been made in line 390, "Three pears (Pyrus x bretschneideri) of uniform size without wounds were selected to test the infection ability of ΔAwHog1 mutant"